# Mixed axial-gravitational anomaly from emergent curved spacetime in nonlinear charge transport

Tobias Holder,[1, 2, *] Daniel Kaplan,[1] Roni Ilan,[2] and Binghai Yan[1, †]

[1]*Department of Condensed Matter Physics, Weizmann Institute of Science, Rehovot, Israel*
[2]*Raymond and Beverly Sackler School of Physics and Astronomy, Tel Aviv University, Tel Aviv, Israel*
(Dated: October 1, 2025)

In 3+1 dimensional spacetime, two vector gauge anomalies are known: The chiral anomaly and the mixed axial-gravitational anomaly. While the former is well documented and tied to the presence of a magnetic field, the latter instead requires a nonzero spacetime curvature, which has made it rather difficult to study. Here, we show that a quantum anomaly arises in the second-order electrical response for zero magnetic field, which creates a dc-current that is both transverse and longitudinal to the electric field. We can identify this new anomaly as a mixed axial-gravitational one, and suggest an experiment in which the anomaly-induced current can be isolated in the second order electrical conductivity. Our findings demonstrate that the semiclassical picture of quasiparticle response needs to be updated: Charge transport generically derives from quasiparticle motion in an emergent curved spacetime, thereby indicating that Fermi liquid theory as it was originally conceived is incomplete beyond linear order.

## I. INTRODUCTION

When cold electrons move in a perfect, static and stable lattice, according to textbook wisdom, they move like almost free quasiparticles in flat space. Disregarding 'dirt'-effects, one might even be tempted to ask, given an excited but low-energy state in vicinity of the Fermi surface, how much does the lattice really matter? Since several years, the impactful insight has been that yes, the lattice matters on a fundamental level due to nontrivial topology present in the band structure [1–3]. However, this insight sidelines the more subtle question how much the lattice matters *locally* and *dynamically*. Why do electronic quasiparticles actually move in flat space, given that they navigate through a dense forest of lattice sites? The answer is simply that quasiparticles do *not* move in a flat space, not even in a perfect and static lattice at zero temperature. In the following we elucidate this statement by demonstrating that a quantum anomaly can arise in the electric, nonlinear conductivity in absence of any thermal gradients or magnetic fields.

Quantum anomalies appear when a classically conserved quantity is not conserved on the quantum level, but broken by quantum fluctuations [4]. They play a central role in the theoretical and experimental understanding of quantum matter, because they imply the existence of non-conserved processes which would be forbidden classically. While anomalies were first discussed in the standard model of elementary particle physics [5, 6], condensed matter systems like Weyl semimetals can provide accessible experimental platforms to study their effects [7, 8]. Weyl semimetals are three-dimensional compounds which feature pairs of massless chiral fermions [9, 10]. In the presence of a magnetic field Weyl fermions exhibit a chiral anomaly, which means that the quasiparticle number of each chirality is not conserved. By now, the chiral anomaly in Weyl semimetals is well understood theoretically [8, 11–13] and confirmed experimentally by a detecting negative

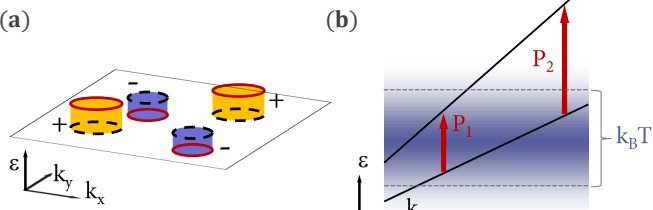

**(a)** **(b)**

Figure 1. (a) Origin of the anomaly according to semiclassics, where all processes are manifest as Fermi surface effects: The applied field renormalizes the dispersion close to the Fermi surface (black, dashed), thereby raising or lowering its energy (red). This leads to an anomalous carrier redistribution between different parts of the Fermi surface, thus violating charge conservation within a given Fermi surface patch (index ±). (b) Origin of the mixed axial-gravitational anomaly according to perturbation theory: Carriers are moved from the Fermi surface into remote bands, thus violating charge conservation at the Fermi level. Such is possible via thermal and non-thermal transitions (blue: thermal window around the chemical potential). At finite temperature, real transitions of type $P_1$ can take place which move carriers from or to the Fermi level. The same process can take place virtually ($P_2$) at second order in the electric field. In both pictures, changes to the occupation lead to a gravitational effect.

magnetoresistance (see Refs. [9, 10, 14] for a review).

Massless chiral fermions additionally exhibit a mixed axial-gravitational anomaly (AGA), which in curved spacetime leads to a non-conservation of the chiral charge and of the energy-momentum tensor [15–17]. Due to its gravitational origin, the AGA is much harder to observe. One approach to probe the AGA is offered in a hydrodynamic picture, where a nonuniform velocity profile can be interchanged for a gravitomagnetic potential, i.e. for an emergent curved spacetime. This is known as the chiral vortical effect [18]. However, in recent years, it became clear that even in the standard Kubo formalism (in flat space) the magnetic field dependence of thermoelectric

transport coefficients is modified by the AGA [18–24]. This finding is reminiscent of Luttinger's suggestion to use a gravitational potential to calculate thermal transport coefficients [25]. However, while thermal effects obviously originate from a modification of the occupation numbers, the conjectured connection of occupation changes with gravitational effects has remained mostly anecdotal [26–28].

Here, we show that changes in occupation are concomitant with the emergence of a non-flat effective spacetime metric. This seemingly surprising finding can be deduced from the appearance of an AGA at second order in the applied electric field, in absence of thermal gradients. However, we emphasize that a relation between second-order responses and occupation changes is not surprising at all on a conceptual level: The second-order electrical conductivity couples quadratically to the electric field, at the same order as Joule heating. These second-order phenomena are both parametrized by vertical (i.e. finite energy) processes, i.e. changes to occupation numbers. While such processes are of thermal nature in case of Joule heating, they are coherent and virtual for the second-order conductivity (Fig. 1). Consequently, if there indeed exists a relation between occupation changes and the emergence of a nontrivial spacetime metric for thermal responses, gravitational effects should likewise appear for non-thermal responses that contain virtual occupation changes. As we demonstrate in the following, semiclassically the same effect can be recovered as a renormalization of the shape of the Fermi surface which vacates or adds carriers in different parts of the Fermi surface (cf. Fig. 1). In the following, the response function of interest is the second order dc-conductivity $\sigma^{(2)}$, corresponding to a static nonlinear current $j_c = \sigma^{(2)}_{ab;c} E_a E_b$ in response to an applied electric field $\boldsymbol{E}$. Using standard perturbation theory, we reveal that $\sigma^{(2)}$ manifests the AGA. Afterwards, the same type of anomaly inflow is demonstrated, which results from the spectral flow between distant Fermi surface patches. As material candidates to observe the anomaly, a certain class of antiferromagnetic Dirac semimetals is suggested, which break both spatial inversion ($P$) and time-reversal symmetry ($T$), while preserving $\mathcal{PT}$. In these materials, conventional intrinsic and extrinsic Hall effects vanish due to $\mathcal{PT}$, making it much easier to isolate the anomaly term.

## II. DIAGRAMMATIC PERTURBATION THEORY

The study of second order conductivity in terms of canonical perturbation theory has a long history [29, 30]. For metallic systems three contributions to $\sigma^{(2)} = \sigma^{dr} + \sigma^{bc} + \sigma^{gr}$ have been described [31–34]. $\sigma^{dr}$ is a purely dispersive term, while $\sigma^{bc}$ is produced by the so-called Berry curvature dipole [32, 35]. In the following, we make use of a new, corrected expression for the third term, $\sigma^{gr}$, and show that it originates from the AGA.

The diagrammatic description of second order optical response has been developed recently [36, 37], which allows to identify the underlying physical processes that create a nonlinear current, and also clarifies the role of finite relaxation rates. Consider a non-interacting fermionic system with band dispersion $\hbar\varepsilon_n(\boldsymbol{k})$ with band index $n$ and the cell-periodic part of the Bloch wavefunctions $|n(\boldsymbol{k})\rangle$, where the lattice momentum $\boldsymbol{k}$ takes values inside the first Brillouin zone. One can then define the complex (Berry) connection $[r_a(\boldsymbol{k})]_{mn} = \langle m(\boldsymbol{k})|\partial_{k_a}|n(\boldsymbol{k})\rangle$ [38] and the Berry curvature $[\Omega_{ab}(\boldsymbol{k})]_{nn} = i\sum_{m\neq n}([r_a(\boldsymbol{k})]_{nm}[r_b(\boldsymbol{k})]_{mn} - [r_b(\boldsymbol{k})]_{nm}[r_a(\boldsymbol{k})]_{mn})$. The Fermi-Dirac function is $f_n(\varepsilon(\boldsymbol{k}))$. When possible, we will henceforth suppress the momentum and band indices.

Evaluating the Kubo formula at second order in the electric field yields six diagrams, which can be rewritten and partially combined by sum rules (cf. App. A). The static limit is then obtained from the optical response by adiabatic switching with $\omega \to i/\tau$, where $\tau$ is a finite lifetime in relaxation time approximation [37, 39]. Taking $\tau \to \infty$, we obtain the three nonlinear conductivities [40]

$$\sigma^{dr}_{ab;c} = -\frac{2e^3}{\hbar^2}\tau^2 \int_{\mathbf{k}} \sum_n f(\partial_{k_a}\partial_{k_b}\partial_{k_c}\varepsilon) \tag{1}$$

$$\sigma^{bc}_{ab;c} = -\frac{2e^3}{\hbar^2}\tau^1 \int_{\mathbf{k}} \sum_n f\left(\partial_{k_a}\Omega_{bc} + \partial_{k_b}\Omega_{ac}\right) \tag{2}$$

$$\sigma^{gr}_{ab;c} = -\frac{2e^3}{\hbar^2}\tau^0 \int_{\mathbf{k}} \sum_n f(\partial_{k_c}G_{ab} + \eta^{tr}_{ab;c}). \tag{3}$$

Here, the Fermi-surface quantity

$$\begin{aligned}&[G_{ab}(\boldsymbol{k})]_{nn}\\&= \sum_{m\neq n} \frac{[r_a(\boldsymbol{k})]_{nm}[r_b(\boldsymbol{k})]_{mn} + [r_b(\boldsymbol{k})]_{nm}[r_a(\boldsymbol{k})]_{mn}}{\varepsilon_m(\boldsymbol{k}) - \varepsilon_n(\boldsymbol{k})},\end{aligned} \tag{4}$$

resembles a momentum-resolved Landau-Zener formula [41, 42]. While Eqs. (1-3) exactly recover the semiclassical expressions for $\sigma^{dr}$ and $\sigma^{bc}$ previously derived in the literature [33], we find a dissimilar expression for $\sigma^{gr}$. Specifically, the transverse part $\eta^{tr}_{ab;c} = \partial_{k_c}G_{ab} - \partial_{k_a}G_{bc}/2 - \partial_{k_b}G_{ac}/2$ of $\sigma^{gr}$ has been first introduced by Ref. [31], whereas Eq. (3) additionally contains the longitudinal component $\partial_{k_c}G_{ab}$. Moreover, only the diagrammatic approach used here gives insight into the permutation symmetry of the spatial indices $(a, b, c)$. Namely, in the dc-limit the quantum effective action and thus gauge-invariant perturbation theory must be invariant under permutations of $(a, b, c)$ [43, 44]. Indeed, from Eqs. (1-3) we conclude that this is the case for $\sigma^{dr}$ and $\sigma^{bc}$ under cyclic permutations, and also for anticyclic ones after recognizing that the operation $a \leftrightarrow b$ implies $\omega \to -\omega$ (i.e. $\tau \to -\tau$). From Eq. (3), it is clear that $\sigma^{gr}$ is not invariant under permutations involving the index $c$ of the current. Such a violation of permutation symmetry means that the current due to $\sigma^{gr}$ cannot be expressed as a functional variation of an effective action. This is a sufficient

criterion [44] to identify $\sigma^{gr}$ as a quantum anomaly [45]. As an additional indicator, we remark that most quantum anomalies originate from triangle diagrams, and indeed, the anomalous nature of $\sigma^{gr}$ originates from the triangle diagram. The identification of a the longitudinal term in $\sigma^{gr}$ as an anomaly at second order constitutes our central result. We now describe several features of this anomaly which identify it as the AGA.

## III. CONTINUITY EQUATION

To shed light on the origin of the anomaly, we now inspect the semiclassical equations of motion. In semiclassics, interband effects like the one expressed in Eq. (4) are not constructed explicitly. Rather, corresponding Fermi surface quantities are introduced which encapsulate their action after projecting to the Fermi surface. The paradigmatic example for this is the Berry curvature, a Fermi surface quantity that is defined using information from the entire band structure [8, 45].

For the collision integral, we make a relaxation time approximation, with the (quantum) lifetime $\tau$ of the quasiparticle. The kinetic equation up to second order in the electric field is known [33],

$$\partial_t f + (\tilde{\boldsymbol{v}} + e\boldsymbol{E} \times \boldsymbol{\Omega}')\partial_{\boldsymbol{r}} f + \frac{e}{\hbar}\boldsymbol{E}\partial_{\boldsymbol{k}} f = -\frac{f - f_0}{\tau} \quad (5)$$

where $f \equiv f(\boldsymbol{k}, \boldsymbol{r}, t)$ is the distribution function, the dispersion contains a shift in energy $\tilde{\varepsilon} = \varepsilon_0 + e^2 G_{ij} E_i E_j$, which also affects the velocity matrix element $\tilde{\boldsymbol{v}}$ and the corrected Berry curvature $\boldsymbol{\Omega}'$. The longitudinal current is defined as

$$\boldsymbol{j} = -e \int_{\boldsymbol{p}} f(\tilde{\boldsymbol{v}} + e\boldsymbol{E} \times \boldsymbol{\Omega}'). \quad (6)$$

At long times, the system relaxes towards the yet to be determined steady-state distribution function $f^{(s)}$. The solution for $f$ up to second order in the electric field is

$$f = f^{(s)} - \frac{\tau e}{\hbar}\boldsymbol{E} \cdot \partial_{\boldsymbol{k}} f_0 + \frac{3\tau^2 e^2}{\hbar^2} E_i E_j \partial_{k_i}\partial_{k_j} f_0. \quad (7)$$

Due to the renormalization of the dispersion, the system will not relax towards the ground state occupation defined by $f_0 \equiv f_\varepsilon$. Instead, adiabaticity demands that at long times the system should occupy states on a shifted Fermi surface according to the occupation function $f_{\tilde{\varepsilon}}$. However, this demand is incompatible with global charge conservation: For a general dispersion, it holds that $\int_{\boldsymbol{k}} f_{\tilde{\varepsilon}} \neq \int_{\boldsymbol{k}} f_0$, which is at odds with Luttinger's theorem. This incompatibility is the source of the anomaly: In the perturbed system, there is a competition between global equilibration within the entire Fermi surface, and the equilibration in the local neighborhood of a given point on the Fermi surface [46]. Since $f_{\tilde{\varepsilon}}$ cannot be the steady-state distribution, the best candidate is almost adiabatic with $f^{(s)} = f^{(0)}_{\tilde{\varepsilon}-\Delta\mu}$, where the chemical potential shift $\Delta\mu$ is

determined by the condition $\int_{\boldsymbol{k}} f^{(s)} = \int_{\boldsymbol{k}} f_0$. The renormalization of the dispersion then induces a persistent and anomalous relaxation dynamics between different Fermi surface patches (FSP). Namely, quasiparticles in patches where $\tilde{\varepsilon} - \varepsilon_0 - \Delta\mu \gtrless 0$ (patch index $\pm$, cf. Fig 1), can relax by decaying to other parts of the Fermi surface with patch index $\mp$. This behavior is reflected in the appearance of anomalous terms in the continuity equations for each patch index. To this end, let us denote the density of states for each patch index by $\nu^{\pm}$. Then, the shift in the chemical potential is given by $\int_{\boldsymbol{k}}(\tilde{\varepsilon} - \varepsilon_0)\partial_\varepsilon f_0 = (\nu^+ + \nu^-)\Delta\mu$, whereas we define for each patch index separately the average $\int_{\boldsymbol{k}\pm}(\tilde{\varepsilon} - \varepsilon_0)\partial_\varepsilon f_0 = \nu^{\pm}\Delta\mu^{\pm}$, which implies $\nu^+\Delta\mu^+ + \nu^-\Delta\mu^- = (\nu^+ + \nu^-)\Delta\mu$. To derive the anomaly-related continuity equation, we then insert the solution to the Boltzmann equation into the kinetic equation, and integrate over the parts of the Fermi surface corresponding to the respective patch index $\pm$, yielding $\partial_t n^\pm + \boldsymbol{\nabla} \cdot \boldsymbol{j}^\pm = -\delta n^\mp/\tau$. Obviously, in the homogeneous bulk it is $\boldsymbol{\nabla} \cdot \boldsymbol{j}^{(i)} = 0$. However, we find for each patch index

$$\partial_t n^\pm = \int_{\boldsymbol{k}\pm} \frac{\partial f^{(s)}}{\partial\varepsilon}\frac{\partial\varepsilon}{\partial t} = \pm\frac{\nu^+\nu^-}{\nu^+ + \nu^-}\frac{\Delta\mu^+ - \Delta\mu^-}{\tau} \quad (8)$$

$$\delta n^\mp = \int_{\boldsymbol{k}\mp}(f^{(s)} - f_0) = \mp\frac{\nu^+\nu^-}{\nu^+ + \nu^-}(\Delta\mu^+ - \Delta\mu^-) \quad (9)$$

Here, in the first line, we made again use of the relaxation time approximation, and used that up to second order in the perturbation it is $f_{\tilde{\varepsilon}-\Delta\mu} \approx f_0 + \partial_\varepsilon f_0(\tilde{\varepsilon} - \varepsilon_0 - \Delta\mu)$. Eqs. (8,9) show that the solution for $f_{\tilde{\varepsilon}-\Delta\mu}$ fulfills the continuity equations in for either patch index, and conserves total charge, i. e. $\partial_t(n^+ + n^-) + \boldsymbol{\nabla} \cdot (\boldsymbol{j}^+ + \boldsymbol{j}^-) = 0$. On the other hand, between Fermi surface patches, we find $\partial_t(n^+ - n^-) + \boldsymbol{\nabla} \cdot (\boldsymbol{j}^+ - \boldsymbol{j}^-) \neq 0$. This means that charge between distant points on the Fermi surface is not conserved at order $\tau^{-1}$, and escapes via the collision term, analogously, but not identically to the mechanism which leads to the chiral anomaly at order $\tau^0$. Also, we reiterate that compared to the latter the anomaly is not produced by the Berry curvature but by $G$ [cf. Eq. (4)]. Since the continuity equation within each FSP is not identically zero, it contains an anomaly, which confirms our previous finding from quantum perturbation theory. The discussion so far was restricted to the steady-state distribution $f^{(s)} = f_{\tilde{\varepsilon}-\Delta\mu}$, which corresponds to uniform scattering between all points on the Fermi surface. This steady state has zero longitudinal current at order $\tau^0$, which is a well known behavior also for the chiral anomaly: The anomaly-related current only emerges if the particle of species $\pm$ decays slowly into the other species $\mp$, otherwise no valley polarization can be built up [47]. In the present case, the latter situation arises when the quasiparticles decay much slower into states in distant FSPs than into states within the same FSP. This is generically true in systems with large Fermi surfaces or multiple Fermi surfaces [48, 49]. Incorporating this modified relaxation dynamics into the kinetic equation, dissimilar steady state distributions arise within each FSP with index $\pm$, i. e.

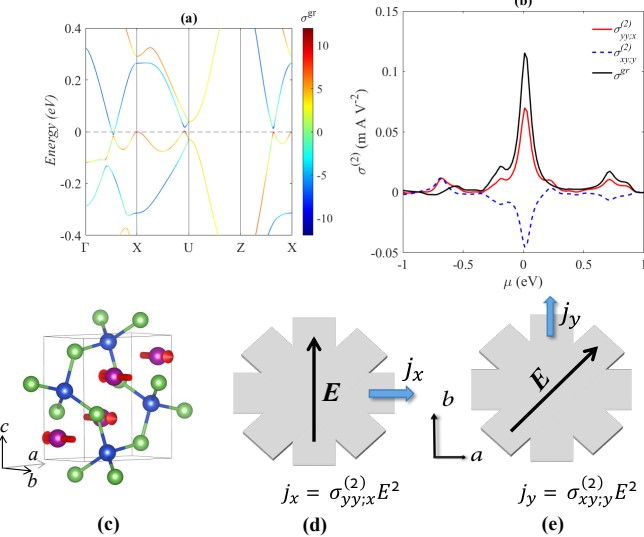

**Figure 2.** Material proposal on the orthorhombic CuMnAs to detect the anomaly. (a) Band structure of CuMnAs in the antiferromagnetic phase with massive Dirac points near the Fermi energy (zero). The color represents the band-resolved nonlinear conductivity due to gravitational anomaly, where the massive Dirac bands contribute significantly. (b) The calculated nonlinear conductivities at different chemical potential ($\mu$). The gravitational anomaly contribution $\sigma^{gr}$ extracted from $\sigma^{(2)}_{yy;x} - \sigma^{(2)}_{xy;y}$. (c) The antiferromagnetic phase of CuMnAs where magnetic moments are along the $b$ axis. (d)-(e) Two experimental setups to measure $\sigma^{(2)}_{yy;x}$ and $\sigma^{(2)}_{xy;y}$ separately. Here, $x, y$ align with the $a, b$ crystalline axes, respectively. In (e), the applied electric field (or bias voltage) orients $45°$ with respect to the $x$ direction. Additionally, the anomaly effect can also be extracted from the scaling of $\sigma^{(2)}$ in Eq. (10).

$(f^{(s)})^{\pm} = f_{\tilde{\varepsilon} - \Delta\mu^{\pm}}$, which then carry the nonzero anomaly-related current $\boldsymbol{j}_{an} = e \sum_{\pm} \Delta\mu^{\pm} \int_{\boldsymbol{k}^{\pm}} \partial_{\varepsilon} f^{(0)} \boldsymbol{v}_0$. We point out that $f^{(s)}$ might acquire an even more complicated structure, depending on the microscopic details [50]. In summary, the term Eq. (3) found in quantum perturbation theory is likewise the source of an anomaly in the semiclassical formalism, and for small scattering between opposite patches, the associated spectral flow between different FSPs leads to a steady-state current.

### A. Origin of the anomaly

We find numerically that in a PT-symmetric material it is $\sigma^{gr} \neq 0$. Thus, the anomaly-related current cannot be created by the chiral anomaly, since the system under consideration is homogeneous, has identical zero Berry curvature at every k-point and there is no magnetic field applied. Therefore, to understand the origin of the anomaly, we recall that gauge invariance under infinitesimal transformations is broken not only by the chiral anomaly, but also by the AGA [43]. The latter has

the form $\epsilon^{\kappa\sigma\alpha\beta} R^{\nu}_{\ \lambda\kappa\sigma} R^{\lambda}_{\ \nu\alpha\beta}$, where $R^{\mu}_{\ \nu\lambda\kappa}$ is the Riemann curvature tensor. Invoking a curvature effect might seem surprising, because there is no curved background geometry. However, we point out that the emergence of an effective nontrivial metric merely encodes the presence of additional accelerations which are not contained in a scalar or vector-like potentials. Indeed, one can show that second order response is sensitive to the (anomalous) accelerations of the quasiparticle [37], which reinforces the notion that the gravitational anomaly will inevitably appear in response functions beyond linear order. We add that the analytical structure of the anomaly in the semi-classical approach is highly indicative af the AGA, which originates from the completely antisymmetric contraction of the Riemann curvature tensor with itself and therefore contains terms with either one or three time derivatives of the spacetime metric. Similarly, the anomaly inflow enters in the semiclassical expression of Eq. (8) through a first time derivative of the density. That the anomaly appears at order $\tau^{-1}$ is thus a very strong indicator for its origin from an emergent curved spacetime.

## IV. EXPERIMENTAL PROPOSAL

Of the nonlinear conductivities $\sigma^{(2)} = \sigma^{dr} + \sigma^{gr} + \sigma^{bc}$, $\sigma^{dr}$ and $\sigma^{gr}$ are nonzero only if the band structure breaks both spatial inversion symmetry ($\mathcal{P}$) and time-reversal ($\mathcal{T}$). Conversely, if the combined symmetry $\mathcal{PT}$ still exists, the Berry curvature contribution $\sigma^{bc}$ vanishes [51]. In materials preserving $\mathcal{PT}$, also the intrinsic linear anomalous Hall effect vanishes and even extrinsic Hall effects due to skew scattering and side jump [52–54] are suppressed [34]. Therefore, $\mathcal{PT}$ materials are ideal candidates to investigate the AGA. On top of that, the two remaining nonlinear Hall contributions, $\sigma^{dr}$ and $\sigma^{gr}$, behave differently under permutation of the spatial indices, so that the Drude contribution $\sigma^{dr}$ can be subtracted away, by taking a combination $\sigma^{(2)}_{yy;x} - \sigma^{(2)}_{xy;y}$. Our proposal for an experimental detection of the AGA is depicted in Fig. 2d-e. It involves a subsequent measurement of $\sigma^{(2)}_{yy;x}$ and $\sigma^{(2)}_{xy;y}$ in the same sample.

We have calculated $\sigma^{(2)}$ and $\sigma^{gr}$ for the orthorohmbic phase of CuMnAs, which is antiferromagnetic and breaks $\mathcal{P}$ and $\mathcal{T}$ but preserves $\mathcal{PT}$ [55, 56]. As shown in Fig. 2a, this material features massive Dirac bands near the Fermi energy ($\mu = 0$) [55, 57]. The second order conductivities $\sigma^{(2)}_{yy;x}$ and $\sigma^{(2)}_{xy;y}$ are shown in Fig. 2c. Although they depend on band structure details, their difference $\sigma^{(2)}_{yy;x} - \sigma^{(2)}_{xy;y}$, the pure AGA contribution, exhibits a peak near the massive Dirac points (Fig. 2c).

As discussed above, the anomaly term $\sigma^{gr}$ is independent of $\tau$, similar to the intrinsic anomalous Hall effect. Thus, we expect $\sigma^{gr}$ to be insensitive to temperature variations or the longitudinal conductivity, even though both $\sigma^{(2)}_{yy;x}$ and $\sigma^{(2)}_{xy;y}$ independently are $\tau$ dependent. In addition, we note that $\sigma^{(2)}$ is sensitive to magnetic or-

dering and the crystal structure of the material. Both $\sigma^{(2)}$ and $\sigma^{gr}$ will vanish if the material recovers inversion symmetry above the Néel temperature.

For an alternative detection scheme, note that in a $\mathcal{PT}$ system, $\sigma^{(2)}$ has two terms which scale differently with the relaxation time $\tau$, in the form of $\sigma^{dr} \propto \tau^2$ and $\sigma^{gr} \propto \tau^0$. Since $\sigma \propto \tau$, we obtain the following relation,

$$\frac{E_\perp}{E_\parallel^2} = \frac{\sigma^{(2)}}{\sigma} = \eta_1 \sigma + \frac{\eta_2}{\sigma}, \tag{10}$$

where $E_\perp$ and $E_\parallel$ are transverse and longitudinal electric field, respectively. Here, $\eta_1 = \frac{\sigma^{dr}}{\sigma^2}$ and $\eta_2 = \sigma^{gr}$. We emphasize that $\eta_2$ directly probes the AGA effect, even without subtracting the permuted $\sigma^{(2)}$ partner. Finally, we emphasize that the scaling of $\sigma^{(2)}$ to the linear conductivity $\sigma$ according to Eq. (10) is different from the Berry-curvature induced nonlinear anomalous Hall effect [58, 59].

To estimate the strength of the nonlinear Hall effect, we adopt the peak value of $\sigma^{(2)}_{yy;x} \approx 0.05$ mAV$^{-2}$ in Fig. 2b, which is in the same order of magnitude as $\sigma^{gr}$. Assuming a reasonable electric field strength $E_y = 1$ V cm$^{-1}$, the effective anomalous Hall conductivity is $\sigma^{AHE} = \sigma^{(2)}_{yy;x} E_y \approx 50$ $\Omega^{-1}\mu$m$^{-1}$, which is comparable to that of some Fe thin-films [60]. The effective Hall angle is then $\gamma = \frac{\sigma^{AHE}}{\sigma} \approx 0.5\%$, where we used $\sigma \approx 10^4$ S cm$^{-1}$ according to Ref. [56].

## V. CONCLUDING REMARKS

We have shown that in systems without time reversal and spatial inversion, an anomalous electrical current appears at second order in the electric field due to the axial-gravitational anomaly. We suggest a straightforward multi-contact geometry to measure this current in an all-electrical setup.

The observations put forth here have quite far-reaching implications for the theory of quantum transport. It allows, for the first time, to probe in the electrical conductivity the nontrivial emergent spacetime in which the electrons move when traversing a periodic lattice. This emergent curved spacetime is consistent with a classical picture of transport of deformable wavepackets flexing and bending while they squeeze through the lattice atoms [61]. These results offer the exciting perspective that dynamical effects of motion in a Riemannian curved spacetime could be engineered and accessed in a genuine bulk setting, unlike it was suggested so far by strain engineering [62–68]. It also opens up a new route to investigate the interplay of anomalies, Bardeen-Zumino currents and their role in quantum field theories in curved spacetime in a lattice, similar to recent developments regarding the chiral anomaly [69]. Finally, the presence of the mixed axial-gravitational anomaly in the nonlinear conductivity implies that Fermi liquid theory as it was originally conceived is incomplete beyond linear order.

## ACKNOWLEDGMENTS

We thank Erez Berg, Tabea Heckenthaler, Johannes S. Hofmann, Karl Landsteiner and Raquel Queiroz for fruitful discussions. B.Y. acknowledges the financial support by the European Research Council (ERC Consolidator Grant No. 815869, "NonlinearTopo") and Israel Science Foundation (ISF No. 2932/21). R.I. was supported by the Israel Science Foundation (ISF No. 1790/18).

Note added: After completion of the manuscript but before publication, a few other works have pointed out the crucial importance of considering the non-flat momentum space geometry for nonlinear response [70–74]. These works support our main conclusions, but are distinct in method and results.

## Appendix

### Appendix A: Derivation of anomalous conductivity from the Kubo formula

The nonlinear conductivity can be directly extracted from the Kubo formalism, by extending the procedure from linear response to account for corrections both to the current operator and the Hamiltonian. Consider a vector potential related to a uniform electric field, $\mathbf{A} = \frac{\mathbf{E}}{i\omega}$. Following minimal coupling, we shift the crystal momentum by $\mathbf{k} \to \mathbf{k} - e\mathbf{A}$. The Hamiltonian and current operators to 2nd order are given by,

$$H \approx H_0(\mathbf{k}) - e\mathbf{v}\mathbf{A} + \frac{e^2}{2}\sum_{a,b} w^{ab} A_a A_b, \tag{A1}$$

$$J^a = -ev^a + e^2 \sum_b w^{ab} A_b + \frac{e^3}{2}\sum_{bc} u^{abc} A_b A_c. \tag{A2}$$

Here we defined the $k$-moments of the Hamiltonian as (all local in momentum space),

$$v_{nm}^a = \left\langle n\mathbf{k} \left| \frac{\partial H_0}{\partial k_a} \right| m\mathbf{k_a} \right\rangle, \tag{A3}$$

$$w_{nm}^{ab} = \left\langle n\mathbf{k} \left| \frac{\partial^2 H_0}{\partial k_a \partial k_b} \right| m\mathbf{k} \right\rangle, \tag{A4}$$

$$u_{nm}^{abc} = \left\langle n\mathbf{k} \left| \frac{\partial^3 H_0}{\partial k_a \partial k_b \partial k_c} \right| m\mathbf{k} \right\rangle. \tag{A5}$$

The frequency $\omega$ introduced above is standard in the $\mathbf{A}$ gauge and will be smoothly taken to zero at the end of the calculation. The full perturbative calculation is carried out through the diagrammatic technique [36, 37, 40]. We briefly summarize the procedure: The expectation values of the current operator Eq. (A2) is given by [75] $j^a = \langle J^A \rangle = -i \int d\Omega \text{Tr}(J^a G^<(\Omega,\omega))$, where the trace acts on band space and the Brillouin zone. $G^<$ is the lesser Green's function obtained from the solution to the Dyson equation. The Dyson equation is given by the series expansion $G^< = \left(\sum_n (\Sigma_r G_r)^n\right) G_0^< \left(\sum_m (\Sigma_a G_a)^n\right)$.

$\Sigma_{r/a}, G_{r/a}$ are the retarded/advanced self-energy and Green's function, respectively, given by $\Sigma_r = -e\mathbf{v}\mathbf{A} + \frac{e^2}{2}\sum_{a,b} w^{ab} A_a A_b + V_{\text{dis}}$, and $V_{\text{dis}}$ is a broadening/disorder contribution evaluated in the relaxation time approximation, i.e., replaced by a constant $\frac{i}{\tau}$. $G_{nm,r/a}(\Omega) = \frac{i\delta_{nm}}{\Omega - \varepsilon \pm \frac{i}{\tau}}$. The unperturbed lesser Green's function is given by $G_0^< = if_n(G_r - G_a)$. We consider a system with finite broadening, and we account for the two distinct poles that appear at 2nd order response through a prescription which ensures adiabaticity [40]. The sum of frequencies pole $\bar\omega = \omega + (-\omega) \to \bar\omega + \frac{2i}{\tau}$, while single frequency poles undergo a shift $\omega \to \omega + \frac{i}{\tau}$. Our results agree with the Boltzmann approach illustrated in the main text in the limit $\tau \to \infty$. To match the two approaches, we resolve the objects in Eq. (A3)-(A5) to their length gauge analogues. The full expansion is given in Ref. [40], but due to PT and crystalline symmetries, we only detail how agreement with the semiclassical formalism is obtained for two cases in the limit $\tau \to \infty$: terms proportional to $\tau^2$ giving the nonlinear Drude weight, and terms proportional to $\tau^0$ giving the anomaly. At order $\tau^2$ the perturbative calculation gives

$$j^c = -\frac{e^3}{\hbar^2}\tau^2 E_a E_b \int_{\mathbf{k}} \sum_n f_n \Big\{ i[r^a, w^{bc}]_{nn} + \frac{i}{2}[r^c, w^{ab}]_{nn}$$
(A6)

$$+ \frac{1}{2}[r^a\Delta^c, r^b]_{nn} - \frac{1}{2}[r^a, \Delta^b r^c]_{nn} - \frac{1}{2}[r^c, \Delta^a r^b]_{nn}$$

$$+ \frac{1}{2}u_{nn}^{abc} - \frac{1}{2}\Big( i[r^b, \varepsilon\tilde\Omega^{ca,0}]_{nn} + i[r^c, \tilde\Omega^{ba,1}]_{nn}$$

$$+ i[r^a, \varepsilon\tilde\Omega^{cb,0}]_{nn} + i[r^c, \tilde\Omega^{ab,1}]_{nn} \Big) \Big\} + (a \leftrightarrow b).$$
(A7)

We introduce the shorthand notation $\tilde\Omega_{nm}^{ab,l} = \big[\varepsilon^l r^a, r^b\big]_{nm}$. For simplicity, band differences such as $f_n - f_m$, and $\varepsilon_n - \varepsilon_m$ are concisely written as $f_{nm}$ and $\varepsilon_{nm}$. The commutator is $[A,B]_{nm} = \sum_l A_{nl}B_{lm} - (a \leftrightarrow b)$. $r^a$ is the Berry connection, as defined in the main text. The reduction of this formula is carried out through Jacobi identities such as,

$$-[r^c, \tilde\Omega^{ba,1}]_{nn} = -[r^b, \varepsilon\tilde\Omega^{ac,0}]_{nn} - [r^a, \tilde\Omega^{bc,1}]_{nn}, \quad \text{(A8)}$$

$$-[r^c, \tilde\Omega^{ab,1}]_{nn} = -[r^a, \varepsilon\tilde\Omega^{bc,0}]_{nn} - [r^b, \tilde\Omega^{ac,1}]_{nn}. \quad \text{(A9)}$$

After inserting the definition of $w^{ab}$ and $u^{abc}$ [40], we arrive at the current expression,

$$j^c = -\frac{e^3}{\hbar^2}\tau^2 E_a E_b \int_{\mathbf{k}} \sum_n f_n \Big( \partial_a \partial_b \partial_c \varepsilon_n \Big). \quad \text{(A10)}$$

In agreement with the semiclassical result. Since $\tau^1$ is expected to vanish, we detail the $\tau^0$ contribution (which constitutes the anomaly). This reads,

$$j^c = \frac{e^3}{\hbar^2} E_a E_b \sum_{nm} \left( f_{nm} \frac{v_{nm}^a w_{mn}^{bc}}{(\varepsilon_{nm})^3} + \sum_l \frac{v_{nm}^a v_{ml}^b v_{ln}^c}{-\varepsilon_{ln}} \left( \frac{f_{nm}}{(-\varepsilon_{mn})^3} + \frac{f_{lm}}{(-\varepsilon_{lm})^3} \right) + (a \leftrightarrow b) \right). \quad \text{(A11)}$$

Introducing the definition of $w^{ab}$ we find two distinct contributions which do not vanish under PT and crystalline symmetries,

$$-\left[\frac{r^a}{\varepsilon^2}, \Delta^b r^c + \Delta^c r^b\right]_{nn} + \left[\frac{r^a}{\varepsilon}, \lambda^{bc}\right]_{nn} - \frac{i}{2}\left[\frac{r^a}{\varepsilon^2}, \tilde\Omega^{bc,1} + \tilde\Omega^{cb,1}\right]_{nn} + (a \leftrightarrow b), \quad \text{(A12)}$$

and,

$$\left[\frac{r^a}{\varepsilon^2}, \Delta^b r^c\right]_{nn} + \left[\frac{r^b}{\varepsilon^2}, \Delta^a r^c\right]_{nn} + i\left[\frac{r^a}{\varepsilon^2}, \tilde\Omega^{bc}\right]_{nn} + i\left[\frac{r^b}{\varepsilon^2}, \tilde\Omega^{ac}\right]_{nn} \quad \text{(A13)}$$

To continue, note that $\varepsilon_{nm}\Omega_{nm}^{ab} = -\tilde\Omega^{ba,1} + \tilde\Omega^{ab,1}$, where $\Omega^{ab}$ is the Berry curvature. Assembling the two contributions gives that $i\left[\frac{r^a}{\varepsilon^2}, \tilde\Omega^{bc} - \tilde\Omega^{cb}\right] = \left[\frac{r^a}{\varepsilon^2}, \varepsilon\Omega^{bc}\right] = -\left[\frac{r^a}{\varepsilon}, \Omega^{bc}\right]$. The Berry curvature is related to the band resolved momentum derivative of the Berry connection via $\lambda^{bc} + \frac{1}{2}\Omega^{cb} = \partial_c r^b$. Importantly, the 2nd order connection $\lambda^{ab}$ [40] appears.

Finally, the current is given by,

$$j^c = \frac{e^3}{\hbar^2} E_a E_b \int_{\mathbf{k}} \left\{ \sum_n f_n \left[ \frac{r^a}{\varepsilon^2}, \Delta^c r^b \right]_{nn} + \left[ \frac{r^a}{\varepsilon}, \lambda^{bc} + \Omega^{cb} \right]_{nn} + (a \leftrightarrow b) \right\} = \frac{e^3}{\hbar^2} E_a E_b \sum_n \int_{\mathbf{k}} \partial_c \left[ \frac{r^a}{\varepsilon}, r^b \right]_{nn}, \quad \text{(A14)}$$

Giving Eq. (3) in the main text.

## Appendix B: Local nature of the perturbation

We prove that Eq. (4) is exact at 2nd order in the applied field, for a uniform electric field. Instead of a applying a direct form of $E$, consider a general perturbation of the form,

$$\hat{V} = V_0(q) e^{iq\hat{r}}. \quad \text{(A1)}$$

The treatment must begin with an expansion of $V$ in Bloch wave functions. We write,

$$V_{nm} = \langle \psi_{nk} | V | \psi_{mk'} \rangle = \int d^d r \psi_{nk}^*(r) V(r) \psi_{mk'}(r). \quad \text{(A2)}$$

Recall that $\langle r | \psi_{nk} \rangle = e^{ikr} u_{nk}(r)$, where $u_{nk}$ is a periodic function such that, $u_{nk}(r) = \sum_G C_{nk}^G e^{iGr}$. $G$ is a reciprocal lattice vector. Note that both $k, k'$ must be held in the first Brillouin zone and all umklapp-like processes are negligible in the low temperature limit. Substituting everything we get,

$$V_{nm}(k, k') = \sum_{G,G'} C_{nk}^{*,G} C_{mk'}^{G'} \int d^d r e^{i(-k+k'+(G'-G))r} V(r) \quad \text{(A3)}$$

Which is a Fourier transform of $V$ at the wavevector $\tilde{q} = (-k + k' + (G' - G))$. Assuming $V(r)$ is a harmonic of $q$, as given,

$$V_{nm}(k, k') = \sum_{G,G'} C_{nk}^{*,G} C_{mk'}^{G'} V_q(q) \delta(q - (-k + k' + (G' - G))). \quad \text{(A4)}$$

Our interest lies in the strict limit $q \to 0$, where a uniform electric field is obtained. Since both $k, k'$ are in the first Brillouin zone, in the $q \to 0$ limit the sum is non-vanishing if and only if $G = G'$. Subsequently, we may rearrange the crystal momenta such that $k' = k + q$. Without altering the sum, we add and subtract the zeroth order q term,

$$V_{nm}(k) = V_q(q) \sum_G C_{nk}^{*,G} \left( C_{m(k+q)}^G - C_{m(k)}^G + C_{m(k)}^G \right)$$

$$= V_q \sum_G C_{nk}^{*,G} C_{mk}^G - iq V_q \sum_G C_{nk}^{*,G} \frac{i \partial C_{mk}^G}{\partial k} + \mathcal{O}(q^2). \quad \text{(A5)}$$

By the orthogonality of the Bloch wavefunctions, the first term is $\propto \delta_{nm}$ and will not appear *in any* off-diagonal expressions such as Eq. (4). To conform with a uniform electric field, we choose $V(q) = \frac{E}{iq}$, and lastly observe that $\sum_G C_{nk}^{*,G} \frac{i \partial C_{mk}^G}{\partial k}$ is exactly $\langle nk | i \frac{\partial}{\partial k} | mk \rangle = \mathbf{r}_{nm}$, as proposed in the main text.

## Appendix C: Probing the anomaly in different crystallographic planes

As stated in the main text, other combinations of the principal axes $x, y, z$ aligned with the $a, b, c$ directions permit measurement of $\sigma^{\text{gr}}$. For completeness, we provide additional calculations, the presence of the anomaly for other spatial indices.

In the event the principle axes are misaligned with respect to the crystallographic directions $a, b, c$, an interpolation is possible through the projection of the directions $\hat{x}, \hat{y}, \hat{z}$. We may define $\cos(\gamma_i) = \hat{r}_i \cdot a_i$, where $a_i = \{a, b, c\}$ as the angle between the principle axes and the crystallographic directions. Next, it is possible to decompose a current in an arbitrary direction, $j^x = \sum_{i,j} A_{ia} A_{jb} A_{xc} \sigma_{ab;c}^{(2)} E_a E_b$. Here $A_{ia}$ is the rotation matrix of the crystallographic direction $a$ about the $i$th principle axis, by the angle $\gamma_i$ defined above. We stress that the choice of coordinate system will not affect the conclusion regarding the presence of an anomaly in the system; a misaligned coordinate system could only *admix* components from different crystallographic directions. When heating the sample above its Neel temperature, we expect the anomalous signal to vanish completely, thus nulling the response in an arbitrary direction as well.

## Appendix D: Computational Methods

Ab initio calculations were performed on orthorhomboic CuMnAs using the full-potential local-orbital minimum-basis code [76]. A $12 \times 12 \times 12$ reciprocal lattice grid was used to obtain the ground state wavefunctions and energies, which were then projected on 144 atomic-site projected Wannier functions. Ground state properties were converged with a tolerance of $10^{-6}$eV for the total energy. All momentum space integrals were carried out on a $350 \times 350 \times 350$ grid in the first Brillouin zone, where convergence was verified when incremental increases in the grid size amounted to less than 5% difference in the integrated value. The anti-ferromagnetic spin configuration of the $Mn$ atoms was implemented in accordance with experimental findings on the commensurate AFM (c-AFM) structure in orthorhombic CuMnAs [55, 56], with spins oriented along the $b$-axis. We recover the experimental result that the c-AFM order breaks the screw symmetry $S_{2z}$, while also breaking the $R_y$ nonsymmorphic reflection symmetry. The symmetry breaking is manifest in the appearance of massive Dirac cones in Fig. 2(a) along the

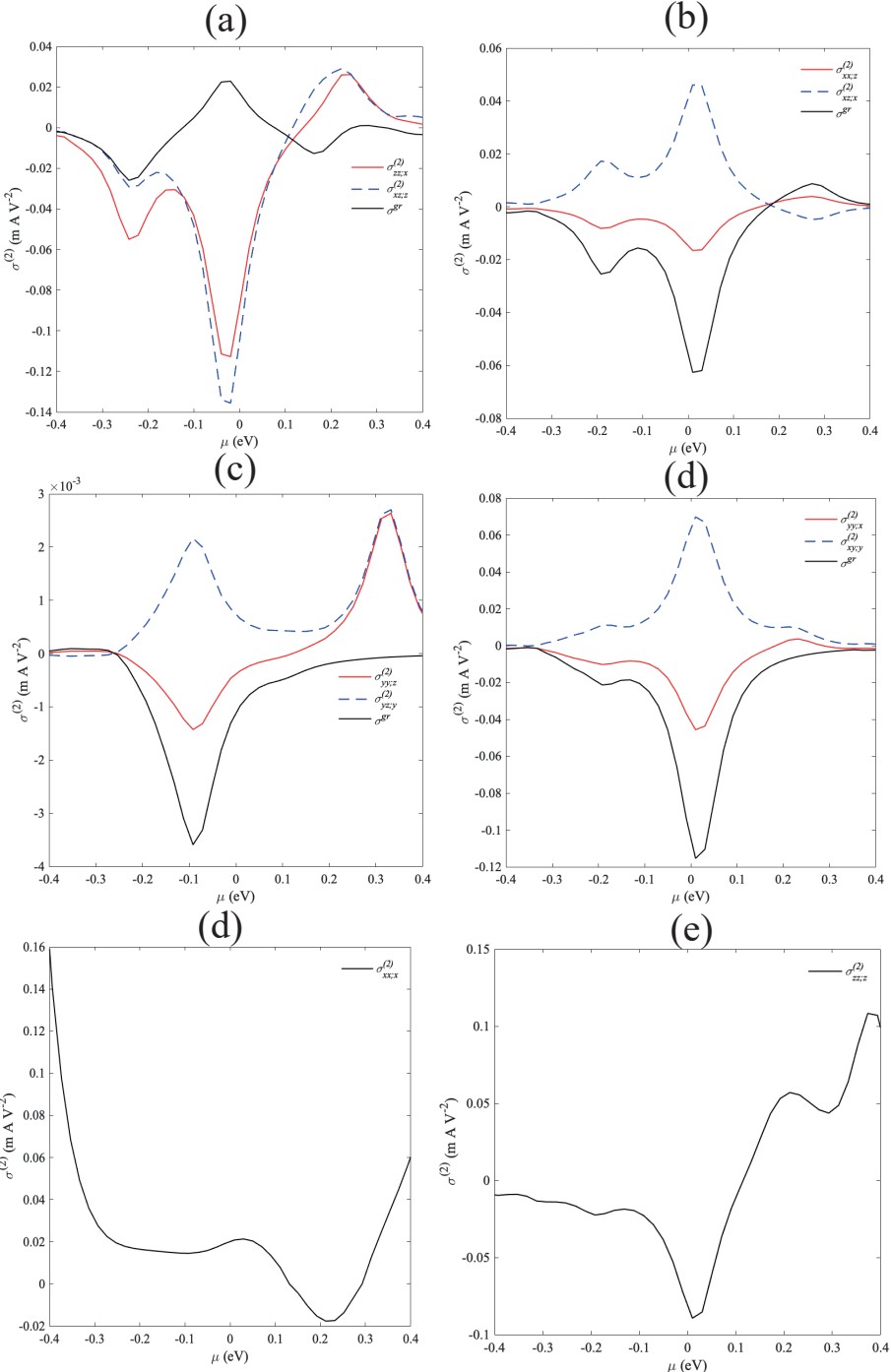

Figure A1. Nonlinear conductivity $\sigma^{ab;c}$ producing non-zero $\sigma^{\mathrm{gr}}$ in the $xy, yz, xz$ planes. The anomaly is generated by the subtraction of $\sigma^{gr} = \sigma^{(2)}_{aa;b} - \sigma^{(2)}_{ba;a}$. We have included longitudinal components $\sigma^{xx;x}$ and $\sigma^{zz;z}$ in (d) and (e), respectively

$\Gamma - X, X - U, Z - X$ lines, unlike the case when spins are aligned with the $c$ direction, and $R_y$ and $S_{2z}$ protect the crossing points [57]. Our observed gap of $\Delta = 5$meV along $\Gamma - X$ is in agreement with previously reported values [55]. The non-magnetic structure of orthorhombic CuMnAs is that of space group 62 (Pnma), which includes the inversion. Upon the introduction of c-AFM order, the space group is reduced to P2$_1$m (11), which breaks the original $S_{2z}$ screw symmetry and $R_y$, as noted above. Besides the conductivities shown in Fig. A1, two other components are symmetry-permitted which do not contribute to the anomaly: $\sigma_{xx;x}^{(2)}$ and $\sigma_{zz;z}^{(2)}$. We note that the longitudinal nonlinear terms can lead to nonreciprocal, unidirectional magnetoresistance in transport due to breaking of both inversion and time-reversal symmetries [77]. Signs of $\sigma_{xx;x}^{(2)}$ and $\sigma_{zz;z}^{(2)}$ can probe the orientation or Neel vector of the antiferromagnetic order.

### 1. Continuity equation and chiral anomaly

For completeness, we reiterate here the procedure used in the main text for the evaluation of the continuity equations, but for the chiral anomaly. We do this to show that firstly, the continuity equations for each particle species are fulfilled by the steady-state distribution, and secondly, the continuity equation for the axial current is nontrivially fulfilled, i. e. the four-divergence of the current is nonzero.

To this end, consider the kinetic equation in the presence of both electric and magnetic field for the distribution function $f^\pm(\boldsymbol{p}, \boldsymbol{r}, t)$ for a given chirality ($\pm$) in relaxation time approximation for the intervalley relaxation rate $1/\tau'$ [8],

$$(1 + \tfrac{e}{c\hbar^2}\boldsymbol{B}\cdot\boldsymbol{\Omega})\partial_t f^\pm + (\boldsymbol{v} + \tfrac{e}{\hbar^2}\boldsymbol{E}\times\boldsymbol{\Omega} + \tfrac{e}{c\hbar^2}(\boldsymbol{\Omega}\cdot\boldsymbol{v})\boldsymbol{B})\partial_{\boldsymbol{r}} f^\pm$$
$$+ (e\boldsymbol{E} + \tfrac{e}{c}\boldsymbol{v}\times\boldsymbol{B} + \tfrac{e^2}{c\hbar^2}(\boldsymbol{E}\cdot\boldsymbol{B})\boldsymbol{\Omega})\partial_{\boldsymbol{p}} f^\pm$$
$$= -(1 + \tfrac{e}{c\hbar^2}\boldsymbol{B}\cdot\boldsymbol{\Omega})\tfrac{f^\pm - f_0^\pm}{\tau'} \qquad (A1)$$

For a given chirality, integrating this over momenta yields the continuity equation, usually stated as

$$\partial_t n^\pm + \boldsymbol{\nabla}\cdot\boldsymbol{j}^\pm \pm \tfrac{e^2}{4\pi^2\hbar^2 c}\boldsymbol{E}\cdot\boldsymbol{B} = -\frac{\delta n^\pm}{\tau'}. \qquad (A2)$$

The distribution function that solves Eq. (A1) is

$$f^\pm = f_0^\pm + \frac{\tau'}{\hbar}\frac{e\boldsymbol{E} + \tfrac{e^2}{c\hbar^2}(\boldsymbol{E}\cdot\boldsymbol{B})\boldsymbol{\Omega}}{1 + \tfrac{e}{c\hbar^2}\boldsymbol{B}\cdot\boldsymbol{\Omega}}\cdot\partial_{\boldsymbol{p}} f^\pm \qquad (A3)$$

This distribution function also fulfills the continuity equation, Eq. (A2), namely $\partial_t n^\pm = 0$, $\boldsymbol{\nabla}\cdot\boldsymbol{j}^\pm = 0$ and $\delta n^\pm = \int_{\boldsymbol{p}}(f^\pm - f_0^\pm) = \pm\tfrac{\tau' e^2}{4\pi^2\hbar^2 c}\boldsymbol{E}\cdot\boldsymbol{B}$. However, due to the anomaly, both sides of Eq. (A2) are actually non-zero, which is a testament to the fact that in the steady state, chiral charge is continuously leaking between chiralities with unequal occupation.

### 2. Absence of equilibrium currents

The effect described in the main text refers to an anomaly current that is created at second order in the electric field. It is thus qualitatively different to the chiral anomaly in several aspects. Most notably, the absence of any magnetic field, and the absence of a Berry curvature at every k-point for materials with $\mathcal{PT}$-symmetry implies that the phase-space measure is not renormalized in the effect described in the main text, and the associated anomaly cannot give rise to any equilibrium currents, very much unlike it was discussed for the chiral magnetic effect [78]. This can be understood intuitively from the fact that in the chiral anomaly, the anomaly current emerges when both an eletric field and a collinear magnetic field are applied. It is then possible to discuss the effect of the electric and magnetic field separately, independently from each other. Importantly, one then needs to worry whether applying only a magnetic field by itself can or cannot lead to a current. This kind of reasoning is not relevant for the electric response at second order, because both fields are in fact the same perturbation, and one cannot selectively turn one of them off.

This is reflected also in the fact that the longitudinal current, irrespective of its origin, is dissipative. Specifically, we find that any current that develops in a metallic system when applying the electric field is dependent on the quasiparticle lifetime through the parts of the conductivity denoted $\sigma_{dr}$ and $\sigma_{bc}$, which are strictly larger than $\sigma_{gr}$ for sufficiently large $\tau$ and thus completely dominate the Joule heating. Therefore, it is neither surprising nor problematic that the longitudinal current at second order is dissipative, and it does not contain any equilibrium currents. In contrast, taking the limit $\tau \to 0$, which seemingly only leaves a current from $\sigma_{gr}$, is inadmissible because it violates the conditions of quasi-adiabaticity under which the response was derived: Any expansion in $\tau^{-1}$ requires that $\hbar\tau^{-1}$ is small compared to the bandwidth. Therefore, the discussion of $\sigma_{gr}$ makes only sense as long as $\sigma_{gr}$ is a small correction to the total conductivity.

### 3. Covariant and consistent formulations of the continuity equations

In the following, we give a more detailed background to the general anomaly equations, and which features are similar and different between the AGA and the more well-known chiral anomaly. We first list the continuity (anomaly-) equations in the so-called covariant formulation, which are then compared to the consistent formulation. We remark that the word 'covariant' in this context has nothing to do with covariant coordinate indices.

Let us first state the general continuity equations in relativistic notation and for a curved spacetime, written in terms of the electromagnetic field strength $F_{\mu\nu}$, the Riemann curvature tensor $R^\mu{}_{\nu\lambda\kappa}$ and the Cristoffel symbol $\Gamma^\mu{}_{\nu\lambda}$. Here, greek indices denote covariant or contravari-

ant spacetime indices, and we use the Einstein summation convention. In this generalized framework, the covariant anomaly equation for the chiral current $J^\mu$ is well known as [43],

$$D_\mu J^\mu_{cov} = \frac{3c_A}{4}\epsilon^{\kappa\sigma\alpha\beta}F_{\kappa\sigma}F_{\alpha\beta} + \frac{c_g}{4}\epsilon^{\kappa\sigma\alpha\beta}R^\nu{}_{\lambda\kappa\sigma}R^\lambda{}_{\nu\alpha\beta} \quad (A4)$$

where $D_\mu$ denotes the covariant derivative, $c_A$ and $c_g$ are the anomaly coefficients of chiral anomaly and AGA, respectively.

$$D_\mu J^\mu_{cov} = \frac{3c_A}{4}\epsilon^{\kappa\sigma\alpha\beta}F_{\kappa\sigma}F_{\alpha\beta} + \frac{c_g}{4}\epsilon^{\kappa\sigma\alpha\beta}R^\nu{}_{\lambda\kappa\sigma}R^\lambda{}_{\nu\alpha\beta} \quad (A5)$$

$$D_\nu T^{\mu\nu}_{cov} = F^\mu{}_\nu J^\nu_{cov} + \frac{c_g}{2}D_\nu(\epsilon^{\kappa\sigma\alpha\beta}F_{\kappa\sigma}R^{\mu\nu}{}_{\alpha\beta}), \quad (A6)$$

The covariant current $J_{cov}$ is $U(1)$-gauge invariant and encodes the Fermi surface contributions to the current. However, this current does not necessarily preserve total charge, even after summing over both chiralities. This non-conservation is found for example in the presence of electromagnetic pseudofields [62, 69, 79, 80]. If this happens, total charge conservation is only restored after the addition of extra currents at the cutoff scale (band edge) [81–83]. The addition of extra currents to $J_{cov}$ results in the consistent current $J_{cons}$, which is not $U(1)$-gauge invariant, but as mentioned does not lose any charge. The latter obeys modified anomaly equations

$$D_\mu J^\mu_{cons} = \frac{c_A}{4}\epsilon^{\kappa\sigma\alpha\beta}F_{\kappa\sigma}F_{\alpha\beta} + (1-\alpha)\frac{c_g}{4}\epsilon^{\kappa\sigma\alpha\beta}R^\nu{}_{\lambda\kappa\sigma}R^\lambda{}_{\nu\alpha\beta} \quad (A7)$$

$$D_\nu T^{\mu\nu}_{cons} = F^\mu{}_\nu J^\nu_{cons} - D_\lambda J^\lambda_{cons}A^\mu$$
$$- \alpha\frac{c_g g^{\mu\nu}}{2\sqrt{-g}}D_\lambda(\sqrt{-g}\epsilon^{\kappa\sigma\alpha\beta}F_{\kappa\sigma}\partial_\alpha\Gamma^\lambda{}_{\nu\beta}), \quad (A8)$$

where $\alpha$ parametrizes a Chern-Simons current $j^\mu_{CS} = \epsilon^{\mu\nu\kappa\sigma}[\Gamma^\lambda{}_{\rho\nu}D_\kappa\Gamma^\rho{}_{\lambda\sigma} + \frac{2}{3}\Gamma^\lambda{}_{\alpha\nu}\Gamma^\alpha{}_{\rho\kappa}\Gamma^\rho{}_{\lambda\sigma}]$ which is added to the action in the form of a contact term $-\alpha\int c_g A\wedge j_{CS}$.

Now, the presence of Fermi sea contributions as given by $j^\mu_{CS}$ depends on the microscopic properties of the system. Let us first consider only the AGA in Eqs. (A5,A7). If band edge contributions are absent, it would be $\alpha = 0$ and thus $J_{cov} = J_{cons}$. Since $J_{cons}$ conserves total charge by construction, and $J_{cov}$ is independent of $\alpha$, the covariant current as induced by the AGA can in principle conserve total charge without Chern-Simons currents. This has to be contrasted with the current if the chiral anomaly terms are nonzero. In the latter case, in Eqs. (A5,A7) it is invariably $J_{cov} \neq J_{cons}$, which means that for the chiral anomaly, additional currents must generically be imposed to restore total charge conservation.

Anomalies are related with a spectral flow in the band structure (i.e. they lead to a redistribution of occupation, cf. [44]). The spectral flow for the AGA is implemented by deforming the band structure, which results in a (local) redistribution of carriers. Importantly, while the total number of carriers is conserved, the number of carriers at the Fermi level in the vicinity of a given k-point is not. In other words, between different patches of the Fermi surface charges are exchanged due to a spectral flow which transfers states from the Fermi sea to the Fermi surface, or vice versa.

* tobiasholder@tauex.tau.ac.il
† binghai.yan@weizmann.ac.il

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
