# Peer review of "Mixed axial-gravitational anomaly from emergent curved spacetime in nonlinear charge transport"

_SciPost Physics_

## Round 1 · Referee Report · Anonymous (Referee 1) · 2025-12-5

Disclosure of Generative AI use

The referee discloses that the following generative AI tools have been used in the preparation of this report:

I used ChatGPT 5.1 to structure the report.

Strengths

1. Conceptual novelty:
The identification of an anomaly-induced nonlinear conductivity term is new and important, expanding the known landscape of nonlinear transport and anomaly physics.
2. Dual theoretical framework:
The authors convincingly combine diagrammatic perturbation theory and semiclassical continuity equations, providing internal consistency and strong support for the anomaly interpretation.
3. Clear experimental direction:
The proposed PT-symmetric CuMnAs platform, with specific conductivity combinations eliminating extrinsic effects, offers a realistic path for detecting the anomaly.

Weaknesses

  1. Emergent metric interpretation is underdeveloped. The manuscript argues that changes in occupation induce an emergent spacetime curvature, but no explicit metric reconstruction is provided. It remains unclear whether this is intended as a literal identification or merely a heuristic analogy.

  2. Dependence on relaxation hierarchy. The semiclassical anomaly relies on the assumption of slow inter-patch and fast intra-patch scattering. This hierarchy may not hold generally, and its material-dependent validity should be discussed.

  3. Consistency of $\tau$-scaling with Fig. 2(b). In Eqs.(1,2,3) the authors organize the second-order conductivity into contributions scaling as $ \sigma^{\mathrm{dr}} \sim \tau^{2}, \qquad \sigma^{\mathrm{bc}} \sim \tau, \qquad \sigma^{\mathrm{gr}} \sim \tau^{0}, $ and argue that only $\sigma^{\mathrm{gr}}$ represents the clean-limit, anomaly-related response in the quasi-clean regime $\tau \rightarrow\infty $. However, in Fig.~2(b) the anomaly contribution $\sigma^{\mathrm{gr}}$ appears comparable in magnitude to the full $\sigma^{(2)}_{yy;x}$. Clarification is needed regarding the $\tau$-regime implicitly assumed in the numerical evaluation of Fig.~2(b), and how the comparability of $\sigma_{\mathrm{gr}}$ and $\sigma^{(2)}$ is consistent with the stated $\tau$-scaling hierarchy.

Report

The manuscript provides a sound theoretical investigation into the role of gravitational anomalies in nonlinear charge transport. The central result that a mixed axial–gravitational anomaly manifests as an intrinsic τ-independent nonlinear conductivity represents a nontrivial extension of anomaly physics into a new transport regime.
The interpretation that virtual transitions generate an effective curvature is intriguing but underspecified. A more concrete or explicit mapping between the microscopic object G_aband an effective metric perturbation would substantially improve conceptual clarity.
Overall, the work is scientifically interesting and addresses a topic at the intersection of topology, geometry, and nonlinear response. The requested revisions below aim to improve clarity, transparency, and accessibility without altering the core scientific contribution.

Requested changes

1. Clarify the emergent curved spacetime interpretation.
Please indicate whether the effective metric is intended as a literal geometric structure or as a heuristic analogy.
2. Discuss the validity and robustness of the patchwise relaxation assumption. Elaborate on when slow inter-patch scattering is expected, how general this assumption is across materials, and how sensitive the anomaly-induced current is to violations of this hierarchy.
3. Clarify the τ-scaling and what is plotted in Fig. 2(b).
Please specify (a) which τ is used, and (b) how this aligns with the clean-limit scaling hierarchy.

Recommendation

Ask for minor revision

---

## Round 1 · Referee Report · Anonymous (Referee 2) · 2025-12-22

Disclosure of Generative AI use

The referee discloses that the following generative AI tools have been used in the preparation of this report:

I declare that ChatGPT 5.2 has been used to structure the report.

Strengths

  1. Predicts a mixed axial-gravitational-anomaly contribution to second-order dc conductivity at zero magnetic field.

  2. Derives a corrected geometric term σ^gr (including a crucial longitudinal contribution) within a Kubo/diagrammatic framework.

  3. Proposes an experimentally relevant system, PT-symmetric, antiferromagnet (CuMnAs) where the proposed mixed axial-gravitational-anomaly contribution can be observed.

Weaknesses

  1. The anomaly claim would be stronger with a clearer statement of the precise conditions under which permutation symmetry must hold in the dc limit and why its violation unambiguously signals a mixed axial-gravitational anomaly.

  2. The section `` Origin of the anomaly'' is too vague in the present form. The authors should be precise in the claims made there.

  3. Although contemporary works on momentum-space geometry are acknowledged, the manuscript should explicitly state what is new in the present manuscript beyond known quantum-geometry contributions to second order nonlinear conductivity, Refs. 70-74. In particular, in Ref. 74, the authors obtain explicit form of the second-order conductivity, analogous to the ones obtained in the present manuscript.

  4. The robustness of the predicted signal from the anomaly against disorder and particular scattering mechanisms beyond a single scattering-time approximation should be addressed.

  5. The CuMnAs case study is discussed in some details, but the impact would increase with a broader set of candidate materials and a short ``design principle'' conditions where a large anomaly contribution can be expected.

  6. The choice of patches on the Fermi surface versus the form of the anomaly is not discussed. In other words, it is unclear how sensitive the anomaly contribution, found here, is on the form of the Fermi surface patches. What does a patch reparametrization imply for the results presented here? Also, it is not clear how the patches have been chosen in the present analysis.

  7. The authors repeatedly used the term ''spacetime". However, I believe the more appropriate terminology would be ''space'' since they are dealing with the only spatial sector defined by the momentum, while the time sector does not explicitly enter. For instance, quantum metric and the tensors derived from it are all defined through the spatial (momentum) derivatives.

Report

The manuscript proposes that a mixed axial-gravitational anomaly can be accessed in PT-symmetric crystalline metals with individually broken time-reversal and inversion symmetries via the second-order (nonlinear) dc conductivity at zero magnetic field. This anomaly emerges from quantum-geometric effects associated with an effective “curved spacetime” description of quasiparticle motion.

To this end, the authors compute the second-order conductivity in the dc limit and decompose it into a nonlinear Drude term, a Berry-curvature-dipole term, and a geometry-related term σ^gr that is τ -independent. They show that σ^gr contains an additional longitudinal piece by using the standard perturbative approach and argue that the full σ^gr violates permutation symmetry required by a gauge-invariant effective-action formulation, which they interpret as a mixed axial-gravitational anomaly. A complementary semiclassical picture is provided in terms of field-induced dispersion renormalization, which in turn generates quasiparticle redistribution on the Fermi surface, and can be interpreted in terms of an effective curved space dynamics, ultimately yielding the anomaly. Finally, antiferromagnetic CuMnAs is proposed as a platform where PT symmetry helps isolate the signal due to the mixed axial-gravitational anomaly.

Recommendation

Ask for major revision

---

## Editorial Decision

awaiting_resubmission